# Gelatin-Based Electrospun Nanofibers Cross-Linked Using Horseradish Peroxidase for Plasmid DNA Delivery

**DOI:** 10.3390/biom12111638

**Published:** 2022-11-04

**Authors:** Kotoko Furuno, Keiichiro Suzuki, Shinji Sakai

**Affiliations:** 1Department of Materials Engineering Science, Graduate School of Engineering Science, Osaka University, 1-3 Machikaneyama-cho, Toyonaka 560-8531, Japan; 2Institute for Advanced Co-Creation Studies, Osaka University, 1-3 Machikaneyama-cho, Toyonaka 560-8531, Japan; 3Graduate School of Frontier Bioscience, Osaka University, 1-3 Yamadaoka, Suita 565-0871, Japan

**Keywords:** gene delivery system, nanofiber, electrospinning, gelatin, horseradish peroxidase

## Abstract

The delivery of nucleic acids is indispensable for tissue engineering and gene therapy. However, the current approaches involving DNA/RNA delivery by systemic and local injections face issues such as clearance, off-target distribution, and tissue damage. In this study, we report plasmid DNA (pDNA) delivery using gelatin electrospun nanofibers obtained through horseradish peroxidase (HRP)-mediated insolubilization. The nanofibers were obtained through the electrospinning of an aqueous solution containing gelatin possessing phenolic hydroxyl (Ph) moieties (Gelatin-Ph) and HRP with subsequent HRP-mediated cross-linking of the Ph moieties by exposure to air containing 16 ppm H_2_O_2_ for 30 min. Then, Lipofectamine/pDNA complexes were immobilized on the nanofibers through immersion in the solution containing the pDNA complexes, resulting in transfection and sustained delivery of pDNA. Cells cultured on the resultant nanofibers expressed genome-editing molecules including Cas9 protein and guide RNA (gRNA), resulting in targeted gene knock-in and knock-out. These results demonstrated the potential of Gelatin-Ph nanofibers obtained through electrospinning and subsequent HRP-mediated cross-linking for gene therapy and tissue regeneration by genome editing.

## 1. Introduction

Delivery of nucleic acids such as DNA and RNA is an attractive technique for tissue engineering [1], wound healing [2], and gene therapy [3]. Nucleic acids modulate the expression of functional proteins to induce tissue regeneration and therapeutic effects [4]. Safe and effective delivery of nucleic acids requires gene delivery systems such as viral and non-viral vectors. Viral vectors show high transduction efficiency; however, they induce immune responses and have limited packaging capacity [5]. In contrast, non-viral vectors based on nanoparticles [6] and polyplexes [7], despite having low transfection efficiency, are suitable as gene delivery systems owing to their low toxicity, the potential for targeted delivery, and relatively low production costs [8]. However, these vectors require systemic administration or local injection, which results in their rapid degradation by nucleases, clearance, off-target distribution, and tissue damage by excessive injections [9].

Scaffolds, such as hydrogels [2,10] and nanofibers [11,12,13,14], with nucleic acids, could be used for gene delivery as an alternative method to eliminate the above issues. These systems are capable of direct and sustained delivery of nucleic acids to cells and tissues [10]. In addition, these scaffolds have other advantages. First, the use of scaffolds with nucleic acids induces long gene expression with reduced toxicity to cells and tissues [15]. Second, local delivery using these scaffolds promotes tissue regeneration by supporting cell adhesion, proliferation, and migration. Third, these scaffolds can protect nucleic acids from extracellular or acidic environments [10,16]. 

Electrospinning is a technique for producing nanofibers by applying high voltage to a polymer solution. Electrospun nanofibers have several advantages, such as an ultrathin diameter and a high surface-to-volume ratio [17]. In addition, they can provide a suitable and beneficial environment for cellular attachment and proliferation owing to their similarity to the natural extracellular matrix [17]. Therefore, they have been applied for tissue engineering, and their combination with nucleic acids has been successfully used for tissue regeneration [18]. 

Herein, we investigated the potential of gelatin-based electrospun nanofibers insolubilized through horseradish peroxidase (HRP)-mediated cross-linking as carriers of genes for gene delivery systems. Gelatin is a natural biomacromolecule derived from collagen and is widely used in the biomedical field owing to its biocompatibility, biodegradability, and low toxicity [19,20]. Additionally, gelatin is water-soluble; therefore, gelatin electrospun nanofibers have been insolubilized using chemical cross-linkers such as glutaraldehyde [21] and carbodiimide [22]. However, glutaraldehyde shows cytotoxicity at high concentrations [21]. Previously, we reported the usefulness of gelatin-based electrospun nanofibers insolubilized through HRP-mediated cross-linking as a substrate for cell culture [23]. HRP catalyzes the cross-linking of phenol and aniline derivatives in the presence of H_2_O_2_ [24], resulting in hydrogel formation from polymers with phenolic hydroxyl (Ph) moieties such as gelatin [25], alginate [26], and hyaluronic acid [27]. 

In this study, we report on gene delivery using electrospun nanofibers of gelatin with Ph moieties (Gelatin-Ph) cross-linked through an HRP-mediated reaction (Figure 1). We obtained the nanofibers from a Gelatin-Ph solution through electrospinning and subsequent HRP-mediated cross-linking by exposure to air containing H_2_O_2_. The cytocompatibility of the resultant nanofibers was evaluated using human embryonic kidney-derived HEK293 cells. To demonstrate gene-delivery potential, we immobilized Lipofectamine/plasmid DNA (pDNA) complexes on the nanofibers and studied their capability of transfection and genome editing of cells.

## 2. Materials and Methods

### 2.1. Materials

Gelatin from porcine skin (type A) and poly(ethylene oxide) (PEO) (Mw = 900,000) were purchased from Sigma-Aldrich Co. LLC (St. Louis, MO, USA). Tyramine hydrochloride (96%) was purchased from Combi-Blocks, Inc. (San Diego, CA, USA). 1-Ethyl-3-(3-dimethylaminopropyl) carbodiimide (EDC) was purchased from Peptide Laboratory Inc. (Osaka, Japan). HRP (190 U/mg), H_2_O_2_, and N-hydroxysuccinimide (NHS) were obtained from FUJIFILM Wako Pure Chemical Co. (Osaka, Japan). Gelatin-Ph (1.95 × 10^−4^ mol-Ph/g) was synthesized through conjugation with tyramine hydrochloride using EDC and NHS as previously described [25]. Propidium iodide (PI), 2-morpholinoethanesulfonic acid, monohydrate (MES), and Cellstain-CytoRed solution (CytoRed) were obtained from Dojindo Molecular Technologies, Inc. (Kumamoto, Japan). Calcein-AM and Dulbecco’s phosphate buffered saline (PBS) were obtained from Nacalai Tesque, Inc. (Kyoto, Japan). Opti-MEM and Lipofectamine 3000 transfection reagents were purchased from Thermo Fisher Scientific, Inc. (Waltham, MA, USA).

### 2.2. Plasmids

Cas9-expression plasmid (hCas9), truncated GFP donor DNA (tGFP), and mTagBFP encoding plasmid (pTRIP-SFFV-mTagBFP-2A) were purchased from Addgene (Addgene 41815, 26864, and 102585, respectively; Watertown, MA, USA). To construct a mTagBFP expression plasmid (pCAX-mTagBFP), the mTagBFP fragment was amplified from pTRIP-SFFV-mTagBFP-2A using PrimeSTAR GXL DNA Polymerase (Takara Bio Inc., Shiga, Japan), and integrated into CAG promoter-derived expression vector (pCAX) using In-Fusion HD Cloning Kit (Takara Bio Inc.). For the knock-in experiment, mutant GFP targeting gRNA expression plasmid (Δg1-gRNA) was generated as previously described [28]. For the knock-out experiment, the GFP target gRNA expression vector (gRNA-GFP) was generated as previously described [11]. 

### 2.3. Cell Lines

Human embryonic kidney-derived HEK293 cells (American Type Culture Collection, Manassas, VA, USA), constitutive GFP-expressing HEK293 cells defined as HEK293EGIPneo [11], and HEK293 cells possessing a 35 bp deletion in the GFP sequence defined as HEK293EGIP*35 cells [28] were cultured in Dulbecco’s modified Eagle medium (FUJIFILM Wako Pure Chemical Co.) containing 10% (*v*/*v*) fetal bovine serum (FBS; Thermo Fisher Scientific Inc.), 1% (*v*/*v*) minimum essential medium non-essential amino acid solution (MEM NEAA; Thermo Fisher Scientific Inc.), and 1% (*v*/*v*) penicillin-streptomycin (Thermo Fisher Scientific, Inc.) in 5% CO_2_ at 37 °C.

### 2.4. Fabrication of Gelatin-Ph Nanofibers through Electrospinning

Gelatin-Ph (12 *w*/*v*%) was dissolved in deionized water at 37 °C and then PEO (1, 2, and 4 *w*/*v*%) was dissolved in the solution at 37 °C. HRP was then dissolved in the solution at the final concentration of 800 U/mL. The polymer solution was placed in a 5 mL glass syringe fitted with an 18-gauge stainless steel blunt needle, and the distance between the tip of the needle and the collector was 7 cm. The solution was electrospun at a flow rate of 0.05–0.08 mm/min under an applied voltage of 18–20 kV using a nanofiber electrospinning unit (Kato Tech Co., Ltd., Kyoto, Japan). Morphology of the nanofibers after gold coating (SC-701 MkII ECO, Sanyu Electron Co., Ltd., Tokyo, Japan) was observed using a scanning electron microscope (SEM; S-2250 N, Hitachi Ltd., Tokyo, Japan). Fiber diameters were measured using ImageJ software v1.53k (Wayne Rasband, Bethesda, MD, USA). The viscosity of these solutions was measured using a rheometer (HAAKE MARS III, Thermo Fisher Scientific, Waltham, MA, USA) equipped with a parallel plate of a 20 mm radius.

### 2.5. Insolubilization of Gelatin-Ph Nanofibers

Gelatin-Ph nanofibers fabricated from the solution containing 2 *w*/*v*% PEO were exposed to air containing H_2_O_2_ and the resulting nanofibers and as-spun nanofibers were degraded in 1 mg/mL collagenase solution (FUJIFILM Wako Pure Chemical Co.), respectively. Fluorescence spectra of the resultant solutions were measured using a fluorescent plate reader (SpectraMax Gemini EM, Molecular Devices, LLC, San Jose, CA, USA). Dityrosine fluorescence was monitored between 360 and 500 nm using an excitation wavelength of 320 nm, with maximum fluorescence intensity around 400–420 nm [29,30]. 

Gelatin-Ph nanofibers prepared from the solution containing 2 *w*/*v*% PEO and HRP were exposed to air containing 16 ppm H_2_O_2_ for 10, 20, and 30 min as previously described [23]. To evaluate the extent of insolubilization, the treated nanofibers were immersed in PBS overnight at 37 °C. 

### 2.6. Cytocompatibility of Gelatin-Ph Nanofibers

The resultant Gelatin-Ph nanofibers (3 mm × 8 mm) were sterilized by UV irradiation for 10 min. HEK293 cells were seeded on the nanofiber mats at a density of 2.5 × 10^3^ cells/mm^2^-mat to evaluate cell viability. After 5 days, the cultured cells were stained with Calcein-AM and PI [25]. Then, they were observed using fluorescence microscopy (BZ-9000, KEYENCE. Co., Ltd., Osaka, Japan), collected by centrifugation at 500× *g* for 5 min, and resuspended in 500 μL of PBS containing 2% (*v*/*v*) FBS for fluorescence analysis using flow cytometry (Beckman Coulter, Inc., Brea, CA, USA). Cell viability was calculated from the numbers of cells stained with Calcein-AM (live) and PI (dead). Cell density was calculated from the cell number measured using a flow cytometer.

### 2.7. Immobilization of Lipofectamine/pDNA Complexes on Gelatin-Ph Nanofibers

Lipofectamine/pDNA complexes were immobilized on the nanofibers by immersion in solutions containing the pDNA complexes, as previously described [11]. Briefly, the pDNA complexes were produced by mixing 1 μg of pDNA with 1 μL of P3000 reagent according to the protocol for Lipofectamine 3000. The nanofibers were immersed in 200 μL of this mixture for 10 min and then rinsed with PBS. 

### 2.8. Transfection Using Gelatin-Ph Nanofibers with Lipofectamine/pDNA Complexes

To evaluate the transfection efficiency and sustained gene delivery of Gelatin-Ph nanofibers with Lipofectamine/pDNA complexes, pCAX-mTagBFP was used. HEK293EGIPneo cells were seeded on the nanofiber mats (5 mm × 7 mm) at a density of 1.4 × 10^3^ cells/mm^2^-mat. To evaluate sustained gene delivery, HEK293EGIPneo cells stained with CytoRed were seeded on the same nanofiber mats at a density of 1.4 × 10^3^ cells/mm^2^-mat after 1 day. Cells were observed using a fluorescent microscope over 4 days of culture. Transfection efficiency was measured using flow cytometry after 4 days of cell culture. Transfection efficiency was determined from the percentage of blue fluorescent protein (BFP)-positive cells, calculated as the number of BFP-positive cells per the number of total cells or CytoRed-stained cells × 100%.

### 2.9. Genome-Editing Using Gelatin-Ph Nanofibers with Lipofectamine/pDNA Complexes

To evaluate the gene knock-in efficiency of the nanofibers with Lipofectamine/pDNA complexes, hCas9, Δg1-gRNA, and tGFP were co-transfected into HEK293EGIP*35 cells. To evaluate gene knock-out efficiency using the nanofibers, hCas9 and gRNA-GFP were co-transfected into HEK293EGIPneo cells. HEK293EGIP*35 or HEK293EGIPneo cells were seeded on the nanofiber mats at a density of 2.9 × 10^3^ cells/mm^2^-mat for gene knock-in or knock-out, respectively. After 7 days of culture, the extent of genome editing was measured using flow cytometry. Genome-editing rates were determined from the percentage of GFP-positive or GFP-negative cells, calculated as the number of GFP-positive or GFP-negative cells per number of total cells ×100%.

## 3. Results and Discussion

### 3.1. Fabrication of Gelatin-Ph Nanofibers

In a previous study, Gelatin-Ph nanofibers were fabricated from solutions of 2–8 *w*/*v*% Gelatin-Ph, 9 *w*/*v*% PEO (Mw = 100,000), 5 *w*/*v*% Pluronic F-127, and HRP through electrospinning as previously described [23]. In this study, we used PEO (Mw = 900,000) for a further enhancement of spinnability based on a previous report describing the effect of the molecular weight of PEO as an additive for an enhancement of spinnability in electrospinning [31]. Gelatin-Ph nanofibers were successfully fabricated from the solutions of 8 *w*/*v*% Gelatin-Ph, 1–4 *w*/*v*% PEO (Mw = 900,000), and HRP through electrospinning (Figure 2a–c). The nanofibers obtained from the solution containing 1 *w*/*v*% PEO had many beads (Figure 2a), which could be caused by the low viscosity of the solution [23,32]. The nanofibers obtained from the solutions containing 2 or 4 *w*/*v*% PEO had no beads (Figure 2b,c). The viscosity of the solutions increased with an increase in PEO concentration (Appendix A) and resulted in the production of nanofibers without beads. The diameter of the nanofibers increased from 235 to 297 nm as the PEO concentration increased from 2 to 4 *w*/*v*%. These results are consistent with previous studies [19,23,33].

### 3.2. Insolubilization of Gelatin-Ph Nanofibers

The Gelatin-Ph nanofibers fabricated from the solution containing 2 *w*/*v*% PEO and HRP were exposed to air containing 16 ppm H_2_O_2_ to insolubilize through HRP-mediated cross-linking of Ph moieties. The formation of dityrosine through the enzymatic reaction was confirmed by the appearance of the fluorescence signal attributed to dityrosine [24,29], which was not observed for the as-spun nanofibers (Appendix A). The nanofibers exposed for 10 and 20 min dissolved immediately in PBS (Appendix A). The exposure to air containing H_2_O_2_ for 10 and 20 min may not be sufficient for H_2_O_2_-induced cross-linking of Ph moieties. In contrast, the nanofibers exposed for 30 min did not dissolve in PBS and cell culture medium (Movie S3). Therefore, in subsequent experiments, we exposed Gelatin-Ph nanofibers to air containing H_2_O_2_ for 30 min. 

### 3.3. Cytocompatibility of Gelatin-Ph Nanofibers

Cytocompatibility of Gelatin-Ph nanofibers was evaluated by measuring the viability and proliferation of HEK293 cells cultured on the nanofiber mats. HEK293 cells were stained with Calcein-AM and PI, which indicated that the cells attached to the nanofiber mats and proliferated (Figure 3a). In addition, cell viability on Gelatin-Ph nanofiber mats obtained from solutions containing 2 and 4 *w*/*v*% PEO was over 90% for 5 days (Figure 3b), which indicated that the amount of PEO in the solution did not affect nanofiber-induced cytotoxicity to HEK293 cells. PEO has previously been reported to have no cytotoxicity [34]. Furthermore, cell density on the nanofiber mats was higher than those on the culture dishes (control) (Figure 3c). This result indicated that the nanofibers promoted cell proliferation as in previous reports [35,36]. In subsequent experiments, we used Gelatin-Ph nanofibers obtained from the solutions containing 2 *w*/*v*% PEO. 

### 3.4. Transfection Using Gelatin-Ph Nanofibers with Lipofectamine/pDNA Complexes

Transfection using Gelatin-Ph nanofibers with Lipofectamine/BFP-expression pDNA complexes was visualized by fluorescent images. Additionally, transfection efficiency in GFP-expressing HEK293 cells (HEK293EGIPneo) cultured on the nanofiber mats was measured by a flow cytometer. HEK293EGIPneo cells expressed BFP for 4 days (Figure 4a–d). To evaluate sustained delivery of Gelatin-Ph nanofibers with the pDNA complexes, the first set of non-labeled HEK293EGIPneo cells was seeded on the Gelatin-Ph nanofiber mats with Lipofectamine/pDNA complexes on day 0. Then, the second set of HEK293EGIPneo cells stained with CytoRed was seeded on the same nanofiber mats on day 1. Fluorescent images indicated that the CytoRed-positive cells seeded later were also transfected (Figure 4b–d). The flow cytometry analysis demonstrated that the total transfection efficiency (i.e., BFP-positive cells) on day 4 was 7.7 ± 2.8% and the transfection efficiency of the later seeded cells (i.e., BFP/CytoRed double-positive cells) was 3.0 ± 1.3%. Therefore, the nanofibers with the pDNA complexes had the ability of sustained delivery of pDNA to cells cultured on them. The pDNA complexes were immobilized on the nanofibers through the electrostatic interaction of cationic pDNA complexes and anionic gelatin [37,38]. Additionally, previous research described that the pDNA complexes lost their ability to deliver pDNA into cells by incubation in a cell culture medium because of the complexation of the pDNA complexes and serum proteins [11,38]. However, immobilization of the pDNA complexes on the nanofibers achieved the sustained delivery of the pDNA complexes. This result indicated that immobilization of the pDNA complexes on the nanofibers inhibited their complexation with serum proteins to effectively deliver pDNA into cells. 

### 3.5. Genome Editing Using Gelatin-Ph Nanofibers with Lipofectamine/pDNA Complexes

The genome editing capability of Gelatin-Ph nanofibers with Lipofectamine/pDNA complexes was evaluated by gene knock-in and knock-out rates measured using previously developed two reporter cell lines, HEK293EGIP*35 and HEK293EGIPneo cells, respectively (Figure 5a) [11]. The gene knock-in and knock-out rates were 0.75 ± 0.20 and 9.7 ± 3.1%, respectively (Figure 5b–d). The nanofibers with the pDNA complexes delivered multiple types of pDNA to the cells cultured on them simultaneously, resulting in gene knock-in and knock-out, which is consistent with previous work [11]. Therefore, the nanofibers with the pDNA complexes can be applicable for gene therapy and tissue regeneration through genome editing. 

## 4. Conclusions

In summary, we developed Gelatin-Ph electrospun nanofibers with Lipofectamine/pDNA complexes for pDNA delivery. Gelatin-Ph electrospun nanofibers were insolubilized by exposure to air containing H_2_O_2_ through HRP-mediated cross-linking. The cross-linked nanofibers showed excellent cytocompatibility. Immobilization of Lipofectamine/pDNA complexes on the nanofibers prevented the complexation of pDNA with serum proteins, which resulted in the sustained delivery of pDNA. Additionally, gene knock-in and knock-out could be achieved using the cross-linked nanofibers. Therefore, Gelatin-Ph electrospun nanofibers with Lipofectamine/pDNA complexes may be a potential candidate for gene delivery and genome editing. 

## Figures and Tables

**Figure 1 biomolecules-12-01638-f001:**
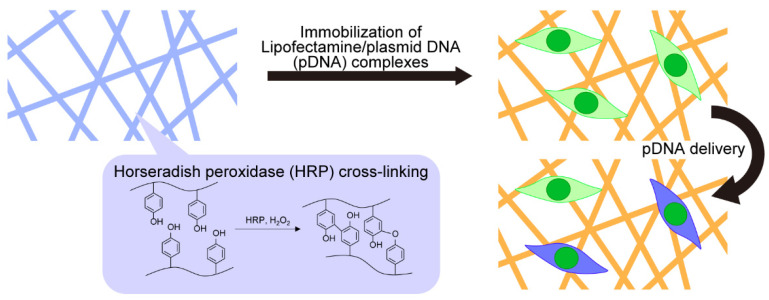
Schematic illustration of insolubilization of nanofibers through horseradish peroxidase (HRP)-mediated cross-linking and immobilization of Lipofectamine/plasmid DNA (pDNA) complexes on the nanofibers with subsequent pDNA delivery to cells cultured on the nanofibers.

**Figure 2 biomolecules-12-01638-f002:**
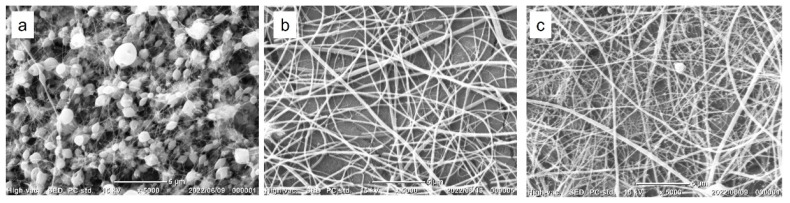
SEM images of Gelatin-Ph nanofibers obtained from 12 *w*/*v*% Gelatin-Ph solutions containing (**a**) 1 *w*/*v*%, (**b**) 2 *w*/*v*%, and (**c**) 4 *w*/*v*% PEO.

**Figure 3 biomolecules-12-01638-f003:**
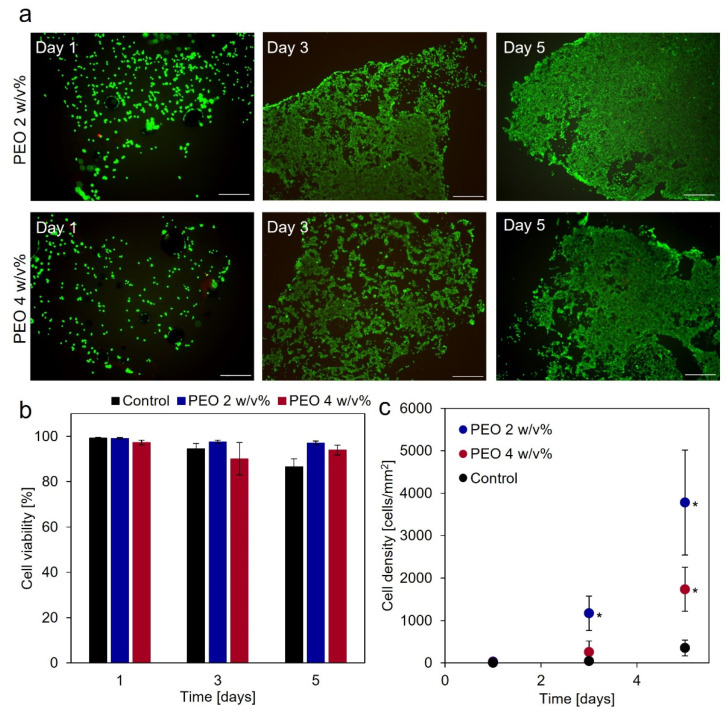
Cytocompatibility of Gelatin-Ph nanofibers obtained from solutions containing 2 and 4 w/v% PEO. (**a**) Fluorescent images of live (green) and dead (red) cells on the nanofibers. Bars are 500 μm. (**b**,**c**) Cell viability and density on the nanofiber mats or the culture dishes (control). The values were measured using a flow cytometer. Data represent mean ± standard deviation (SD; *n* = 5). * *p* < 0.05 compared to control.

**Figure 4 biomolecules-12-01638-f004:**
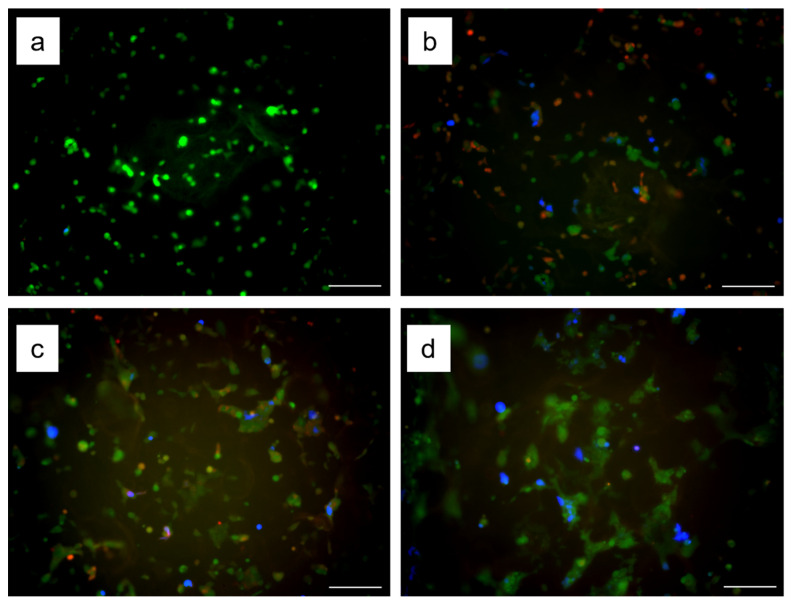
Time course of transfection by Gelatin-Ph nanofibers with Lipofectamine/pDNA complexes. The fluorescent images of cells (green), cells seeded later (red), and transfected cells (blue) on days (**a**) 1, (**b**) 2, (**c**) 3, and (**d**) 4 after initial cell seeding. Bars are 200 μm.

**Figure 5 biomolecules-12-01638-f005:**
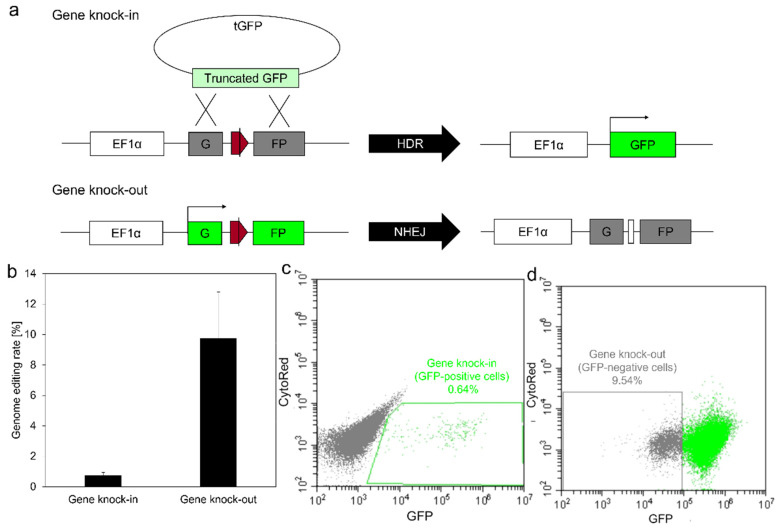
Genome editing using Gelatin-Ph nanofibers with Lipofectamine/pDNA complexes. (**a**) Schematic representation of genome editing reporter assays for gene knock-in and knock-out. EF1α promoter drives GFP or mutated GFP. Red pentagon, Cas9/gRNA target sequence. Black line within the red pentagon, Cas9 cleavage site. tGFP, HDR-donor carrying truncated GFP. HDR, Homology-directed repair. NHEJ, Non-homologous end joining. (**b**) Genome editing rates of gene knock-in and knock-out. Data represent mean ± SD (*n* = 5). (**c**,**d**) Representative flow cytometry images showing the gene knock-in (**c**) and knock-out (**d**) rates.

## Data Availability

All data generated or analyzed during this study are included in this published article and its Appendix A.

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
