# Peer review of "Gelatin-Based Electrospun Nanofibers Cross-Linked Using Horseradish Peroxidase for Plasmid DNA Delivery"

_biomolecules, 2022, doi:10.3390/biom12111638_

Round 1
Reviewer 1 Report
1. What is the PEO content of Gelatin-Ph nanofibers in the insolubilization experiment? 2 w/v% or 4w/v%?
2. It should provide experimental evidence for the insolubility of Gelatin-Ph nanofibers, such as real-time video.
3. In addition to insoluble characterization, the authors should provide other phy-chemical characterization, such as morphology, swelling, FTIR, etc.
4. It says "The number of HEK293EGIPneo cells expressing BFP increased over days" , but the expression of BFP in figure 4d was not the highest. Please confirm it.
Reviewer 2 Report
Review on manuscript biomolecules-1989147
The article is about the presentation of the plasmid DNA delivery using gelatin electrospun nanofibers obtained through horseradish peroxidase-mediated insolubilization.
The article is well-written and the rsults are worth to be publish.
I have some minor comments:
- in the Introduction the citation of some references containing similar results must be done: e.g. 10.1002/mabi.202000040
- the purity of the applied chemicals is missing. (line 79-82)
- the description of the measuring techniques must be improved.
- Fig. S1 contains rheology results. The concentration must be presented in the figure legend.
Round 2
Reviewer 1 Report
I think the present form can be accepted.